# Structural Insights into the Mechanisms Underlying Polyaminopathies

**DOI:** 10.3390/ijms25126340

**Published:** 2024-06-07

**Authors:** Bing Wu, Sen Liu

**Affiliations:** 1Cooperative Innovation Center of Industrial Fermentation (Ministry of Education & Hubei Province), Key Laboratory of Fermentation Engineering (Ministry of Education), Wuhan 430068, China; 2Hubei Key Laboratory of Industrial Microbiology, National “111” Center for Cellular Regulation and Molecular Pharmaceutics, Hubei University of Technology, Wuhan 430068, China

**Keywords:** polyamine, Snyder–Robinson syndrome, Bachmann–Bupp syndrome, deoxyhypusine synthase disorder, Faundes–Banka syndrome, deoxyhypusine hydroxylase disorder

## Abstract

Polyamines are ubiquitous in almost all biological entities and involved in various crucial physiological processes. They are also closely associated with the onset and progression of many diseases. Polyaminopathies are a group of rare genetic disorders caused by alterations in the function of proteins within the polyamine metabolism network. Although the identified polyaminopathies are all rare diseases at present, they are genetically heritable, rendering high risks not only to the carriers but also to their descendants. Meanwhile, more polyaminopathic patients might be discovered with the increasing accessibility of gene sequencing. This review aims to provide a comprehensive overview of the structural variations of mutated proteins in current polyaminopathies, in addition to their causative genes, types of mutations, clinical symptoms, and therapeutic approaches. We focus on analyzing how alterations in protein structure lead to protein dysfunction, thereby facilitating the onset of diseases. We hope this review will offer valuable insights and references for the future clinical diagnosis and precision treatment of polyaminopathies.

## 1. Introduction

Polyamines are a class of organic polycations that exist in nearly all living organisms. In mammalian cells, the major polyamines are putrescine, spermidine, and spermine [1]. Polyamines contain positive charges under the physiological pH value (7.4), so they can bind to negatively charged components, such as DNA, RNA, proteins, and membranes. Through these interactions, polyamines are involved in various biological processes, including DNA replication, RNA transcription, and protein translation, affecting cell fates ranging from growth to proliferation, differentiation, and apoptosis [1]. Echoing the important biological functions of polyamines, their intracellular level is stringently maintained by polyamine synthesis, catabolism, and transport (Figure 1). The major proteins in the biosynthesis pathway of polyamines are ornithine decarboxylase (ODC or ODC1), spermidine synthase (SRM), spermine synthase (SMS), and S-adenosylmethionine decarboxylase 1 (AdoMetDC or AMD1). The catabolism pathway is mainly regulated by spermine oxidase (SMOX), spermidine/spermine N1-acetyltransferase (SSAT1), and polyamine oxidase (PAOX). The membrane transport of polyamines is mediated by a range of membrane transporters, with SLC3A2 [2] and ATP13A3 [3] being the best-known ones.

Whereas cellular polyamine homeostasis is sophistically maintained by the polyamine metabolism network, the utilization of polyamines remains to be thoroughly investigated. Currently, one of the well-elucidated roles of polyamines is the involvement of spermidine in the hypusine modification (hypusination) of the eukaryotic translation initiation factor 5A (EIF5A), which converts the inactive EIF5A precursor into the active form (Figure 1). In the first step of EIF5A hypusination, deoxyhypusine synthase (DHPS) cleaves out the 4-aminobutyl group from spermidine and transfers it to the ε-amino group of the lysine residue K50 in EIF5A, generating a deoxyhypusine intermediate. In the second step, the deoxyhypusine is hydroxylated by deoxyhypusine hydroxylase (DOHH) to form hypusine, leading to the activation of EIF5A. The hypusinated EIF5A then binds to the ribosome to help with the translation of proteins containing polyproline sequences [4,5]. Since the expression of over 400 proteins might be regulated by EIF5A [6], it is not surprising that the hypusination of EIF5A is highly conserved.

For a long time, whether and how polyamine homeostasis directly causes diseases has been controversial. Excessive depletion or accumulation of polyamines can negatively affect cells in aging [7], renal failure [8], neurodegenerative disease [9], and cancer [10]. Elevated levels of polyamines can promote the proliferation of cancer cells and tumor growth, while decreased levels of polyamines may lead to cell growth arrest. For example, polyamine levels are significantly elevated in colon cancer, so polyamines could serve as potential biomarkers for colon cancer [11]. Altered EIF5A function and hypusination have also been associated with various human diseases, including diabetes [12], viral infection [13], neurodegenerative disease [14], and cancer [15]. For example, EIF5A can promote the proliferation and survival of cancer cells by regulating protein synthesis and modulating autophagy and apoptosis, thereby driving cancer initiation and progression [5]. More distinctly, the homozygous whole-body deletion of any of the genes of ODC [16], AMD1 [17], EIF5A [18], DHPS [18], or DOHH [19] in mice results in embryonic lethality. However, these studies have not established the causative role of polyamine metabolism in specific diseases. The direct connection between polyamines and diseases becomes even more elusive after failures pile up in the clinical trials of ODC inhibitors, AdoMetDC inhibitors, and polyamine mimics.

In recent years, studies supported by molecular details have accumulated to establish the causative role of polyamine metabolism in specific diseases (Figure 1). In 2003, Cason et al. reported the first case of Snyder–Robinson syndrome (SRS), a rare genetic disease caused by mutations in SMS [20]. In 2018, Bachmann et al. identified mutations in ODC leading to the rare genetic disorder known as Bachmann–Bupp syndrome (BABS) [21]. More diseases have been linked to polyamine utilization. In 2019, Ganapathi et al. revealed that DHPS mutants with reduced enzyme activity might be associated with a neurodevelopmental disorder [22]. In 2021, Faundes et al. established that dysfunctional EIF5A variants cause Faundes–Banka syndrome (FABAS) [23]. In 2022, Ziegler et al. found that missense and truncating DOHH variants were associated with a neurodevelopmental phenotype [24]. Enlightened by these important discoveries in the polyamine research field, the term polyaminopathies was coined to refer to these conditions [25]. In this paper, we provide a comprehensive review of structural insights into the mutated proteins in the reported polyaminopathies (Table 1 and Table 2).

## 2. Snyder–Robinson Syndrome

Snyder–Robinson syndrome (SRS) is the first X-linked syndrome linked to the dysregulation of polyamine homeostasis and was first reported by Snyder and Robinson in 1969 [26]. The genetic cause of this syndrome was not established until 2003, when Cason et al. demonstrated that this syndrome was caused by mutations in SMS [20]. Clinically, female carriers are normal, whereas male carriers show varying degrees of developmental disabilities, such as intellectual disability, developmental delay, hypotonia, weakness, seizures, osteoporosis, and kyphosis, as well as walking anomalies, facial deformities, genital abnormalities, and renal complications. Thus far, 24 patients have been reported, among which 1 died of hypoxic–ischemic encephalopathy at the age of 4 and another died of secondary sepsis at 4 months old.

SRS is caused by the partial or complete loss of SMS’s enzyme activity, which leads to decreased putrescine/spermine levels and elevated spermidine levels. Spermidine accumulation might increase the production of toxic aldehydes and reactive oxygen species, disrupting lysosomal and mitochondrial functions. SMS is a homodimeric enzyme, and the monomer consists of three structural domains: the N-terminal domain associated with dimerization, the C-terminal domain containing the active site, and the central domain [27] (Figure 2). Among the 13 reported clinical SMS mutants, 5 contain mutations located in the N-terminal domain. Among them, p.M35R [28] significantly destabilizes the monomer structure, but p.G56S [29], p.F58L [30], p.G67E [31], and p.P112L [28] reduce both the monomer stability and the dimerization affinity. Two mutations, p.R130C and p.V132G, are located in the loop between the N-terminal domain and the central domain. p.R130C [32] may destabilize the dimer as well as affect the structure of the neighboring substrate-binding site. p.V132G [33] significantly reduces the activity of SMS. This mutation might also affect the conformation of the loop region and the active site. p.Q148R and p.I150T are located in the central domain. p.Q148R [34] alters the 5′-methylthioadenosine (MTA)-binding site, whereas p.I150T [35] induces structural changes in the vicinity of the MTA-binding site and reduces the stability of the C-terminal domain. p.L277F, p.M303Kfs*3, and p.Y328C are located in the C-terminal domain. p.L277F [36] alters the MTA- and substrate-binding sites. p.M303Kfs*3 [37] introduces a termination codon prematurely, which may trigger nonsense-mediated mRNA degradation. Accordingly, the protein is completely undetectable by Western Blotting [37]. p.Y328C [38] elicits a strong effect on the conformational dynamics and hydrogen-bonding network around the active site. The c.329+5G>A [20] splicing abnormality results in the deletion of twenty-two amino acids from the fourth exon of the gene. However, experiments proved that this mutant was still able to be correctly spliced on a small scale to synthesize some normal SMS proteins.

Considering that the SMS variants in SRS patients lose partial or all enzyme activity, the major therapeutic strategy is rebalancing the spermidine/spermine ratio [39]. Tantak et al. designed a prodrug containing a redox-sensitive quinone “trigger”, a “trimethyl lock” aryl “release mechanism”, and spermine [40]. This prodrug selectively delivered spermine into fibroblast cells and showed significant beneficial effects in the cells of patients with inactive SMS variants. Tao et al. suggested that SSAT1 may be a potential treatment target for SRS [41]. SSAT1 is the rate-limiting enzyme in the polyamine catabolism pathway, using acetyl-CoA as an acetyl donor to acetylate polyamines. Sodium phenylbutyrate (PBA) can be efficiently catabolized in vivo to phenylacetylcoenzyme A, and then the latter can competitively inhibit SSAT1 activity, restoring the acetyl coenzyme A level and reducing the spermidine level in fibroblast cells from SRS patients [41]. Stewart et al. noticed that (R,R)-1,12-dimethylspermine (Me_2_SPM) caused a significant decrease in spermidine content in SRS patients, but the mechanism of action is not clear [42]. They also found that DFMO reduced spermidine biosynthesis and increased spermine content by stimulating spermine synthesis and uptake. They further discovered that the combination of DFMO and Me_2_SPM reduced spermidine and total polyamines in the cells of SRS patients [43].

## 3. Bachmann–Bupp Syndrome

Bachmann-Bupp syndrome (BABS) is the second disease proven to be associated with the dysregulation of polyamine metabolism. BABS is a rare neurodevelopmental disorder caused by mutations in the C-terminal sequence of ODC and was first reported by Bupp et al. in 2018 [21]. The clinical features of BABS patients are intellectual disability, developmental delay, hypotonia, non-congenital alopecia, non-specific brain MRI abnormalities, and non-specific malformations, along with macrocephaly, minor facial deformity, prenatal amniotic fluid excess, and hyperopia. Thus far, 11 patients have been reported [21,44,45,46]. One case was a male stillborn with an abnormal fetal MRI at 34 weeks of gestation [44].

ODC is a 5’-pyridoxal phosphate (PLP)-dependent homodimeric enzyme that catalyzes the decarboxylation of ornithine to produce putrescine. The ODC’s enzyme activity can be inhibited by the binding of ODC antizyme 1 (OAZ1), which induces the exposure of the 37 C-terminal residues (PEST degron) of ODC, leading to the ubiquitin-independent degradation of ODC by the 26S proteasome [47]. In BABS patients, mutations in the PEST degron cause partial or total loss of the 37 residues (Figure 3), resulting in its inability to be recognized and degraded by the 26S proteasome. At the same time, the truncated ODC remains catalytically active and more stable, leading to elevated cellular levels of putrescine and acetylputrescine [25].

DFMO is a specific, irreversible ODC inhibitor. BABS patients treated with DFMO gained normal hair growth, improved muscle tone, and recovered physical development without significant side effects [48].

## 4. Deoxyhypusine Synthase Disorder

Deoxyhypusine synthase disorder (DHPS disorder) is a neurodevelopmental disorder caused by biallelic mutations in DHPS and was first reported by Ganapathi et al. in 2019 [22]. The clinical features of patients include mental retardation, developmental delay, seizures, dystonic abnormalities, and pregnancy problems (gestational hypertension, pre-eclampsia, HELLP syndrome, oligohydramnios), along with walking difficulty and mild facial deformities. Thus far, five patients from four families have been reported.

DHPS is a tetrameric enzyme consisting of four identical subunits [49]. DHPS uses nicotinamide adenine dinucleotide (NAD) as a cofactor and catalyzes the first step of EIF5A hypusination [50] (Figure 4). Both NAD and the substrate, spermidine, bind in the catalytic pocket located at the interface of the two DHPS subunits. The DHPS disorder is caused by DHPS mutants with decreased enzyme activity. These mutants lead to a decrease in hypusinated EIF5A (EIF5A^Hyp^) in the cytoplasm and an increase in acetylated EIF5A (EIF5A^Ack47^) in the nucleus. This change, in turn, leads to impaired mRNA translation and alters the synthesis of proteins involved in neurodevelopment [22]. Among the reported mutations, p.N173S is shared by all known DHPS disorder patients. p.N173S fails to form hydrogen bonds with neighboring amino acids (V17, K19), which reduces the binding affinity of spermidine. It also affects the correct docking and interaction of EIF5A in the active site of DHPS [51]. p.Y305_I306del leads to the failure of spermidine binding and may also attenuate the binding of NAD. Moreover, this mutation destabilizes the structure of DHPS, and the mutated DHPS dimer cannot form tetramers. p.Y305A alone showed reduced binding affinity to spermidine and NAD, but its enzyme activity was normal [52]. In in vitro experiments, the p.N173S mutant had around 20% of the enzyme activity compared to the wild-type DHPS, while the p.Y305_I306del mutant completely lost its enzyme activity. p.Met1? alters the start codon of translation, leading to the failure of DHPS synthesis. The c.1014+1G>A splicing abnormality leads to the deletion of two exons at the C-terminal region of DHPS, within which K329, a key residue in the active site, is located. Therefore, this variant leads to a reduction or loss of enzyme activity. Ganapathi et al. suggested that a complete DHPS deficiency could be incompatible with normal human embryonic development [22].

## 5. Faundes–Banka Syndrome

Faundes-Banka syndrome (FABAS) is a neurodevelopmental disorder caused by heterozygous EIF5A variants and was first reported by Faundes et al. in 2021 [23]. The clinical features of patients are mental retardation, developmental delay, facial dysmorphisms, and microcephaly, along with hypotonia and cardiac abnormalities. Thus far, seven patients have been reported.

EIF5A is a small protein consisting of two β-sheet domains (Figure 5). The alkaline N-terminal domain contains the hypusination site [53,54]. The acidic C-terminal domain resembles an oligonucleotide-binding fold. The sequence homology between human EIF5A and yeast EIF5A is as high as 64% [55], and human EIF5A can be a substitute for yeast EIF5A in protein function [56]. In the yeast model, deleting either the N-terminal domain or the C-terminal domain of human EIF5A was lethal [54]. In FABAS, mutations in EIF5A lead to reduced interaction of EIF5A with the ribosome and decreased synthesis of polyproline-containing proteins [23]. p.T48N is close to the K50 hypusination site and hinders EIF5A hypusination. The arginine residue R109 has various mutations, including the missense mutation p.R109G, the nonsense mutation p.R109*, and the frameshift mutation p.R109Tfs*8. This amino acid is encoded by the CGA codon, and the CG dinucleotide is a mutation hotspot in human diseases because it is prone to methylation and deamination. In the computer-simulated complex structure of yeast EIF5A and the 60S ribosome, G106 and R109 are close to the uL1 ribosomal protein, and E122 is close to the P-site tRNA [23].

Spermidine supplements partially restored EIF5A function and its resultant phenotypes in the yeast model and the zebrafish EIF5A transient knockdown model [23]. In yeast, spermidine partially or entirely restored the interaction of EIF5A with the 80S ribosome, although the content of EIF5A and the level of hypusination were not enhanced. The mechanism by which spermidine enhances the interaction of EIF5A with the 80S ribosome remains unclear.

## 6. Deoxyhypusine Hydroxylase Disorder

Deoxyhypusine hydroxylase disorder (DOHH disorder) is a neurodevelopmental disorder caused by a biallelic mutation in DOHH and was first reported by Ziegler et al. in 2022 [24]. The clinical features of patients are intellectual disability, developmental delay, microcephaly, congenital heart malformations, and brain MRI abnormalities, along with visual impairment and facial deformities. Thus far, five patients from four families have been reported. One female patient died of heart failure at the age of 15. One male patient died of multi-organ failure due to a severe lung infection at 25 months old.

DOHH is a unique non-heme di-iron monooxygenase. The DOHH structure has a symmetrical dyad composed of eight HEAT-repeat domains in tandem (Figure 6). DOHH contains metal coordination sites (HE sites) consisting of four highly conserved histidine–glutamate motifs, namely H56-E57, H89-E90, H207-E208, and H240-E241. These HE sites are essential for enzyme activity and are involved in the binding of Fe^2+^ and substrates [57,58]. The DOHH disorder is caused by DOHH mutants with decreased enzyme activity. These mutants result in the accumulation of deoxyhypusine-containing EIF5A (EIF5A^Dhp^), the decrease in EIF5A^Hyp^, and impaired mRNA translation [24]. p.G102Kfs*6 loses the last two HE sites and has a very low expression level, possibly triggering nonsense-mediated mRNA decay. p.G219Nfs*54 loses the last HE site. These two mutants are inactive in in vitro activity assays. p.Y280* loses the C-terminal 22 amino acids and has low enzyme activity. In vitro experiments showed that p.P152L, p.N184K, and p.I249T had enzyme activities similar to the activity of the wild-type DOHH, but the enzyme activities of these mutants in patients were significantly lower compared to the wild-type protein in healthy individuals. This discrepancy will need further investigation.

## 7. Conclusions and Perspectives

Polyamines play vital roles in a wide range of cellular processes. Spermine was first discovered by Antonie van Leeuwenhoek in 1678 [59], and most studies in the polyamine research field have been focused on inhibiting the upregulated polyamine level in cancers. In recent years, polyamines have been linked to healthy aging, and their intake has been found to positively impact health maintenance and the control of various diseases by inducing autophagy and inhibiting necrosis in mice [60], yeast [61], and human cells [62]. These studies undoubtedly indicate that maintaining cellular polyamine homeostasis is crucial for health. The polyaminopathies reviewed in this paper strongly support the causative effect of the dysregulation of polyamine homeostasis and the polyamine metabolism network in diseases. More encouragingly, the ODC inhibitor DFMO was approved in December 2023 by the FDA to reduce the risk of relapse in adult and pediatric patients with high-risk neuroblastoma.

Although the five polyaminopathies discovered thus far are different in their mutated genes, they have many aspects in common. Firstly, they are all rare diseases. This might be due to the low clinical detection rate, limited by the availability and cost of gene sequencing. Alternatively, this might indicate that the polyamine metabolism network is too crucial to be perturbed for viability. Secondly, clinical features are very similar in polyaminopathies. All clinical patients had neurodevelopmental disorders and bone abnormalities, proving that their pathogenic mechanisms have a common origin. Both BABS and SRS cause polyamine catabolic abnormalities, such as an increase in the SSAT1 level. SSAT1 acetylates spermidine and spermine, thereby increasing their extracellular transport and oxidative decomposition by PAOX. SSAT1 is also able to inactivate EIF5A by acetylating hypusinated EIF5A [63].

Very recently, Alayoubi et al. reported spermidine/spermine N1-acetyltransferase-like 1 (SATL1) gene mutations in two male patients with autism spectrum disorder (ASD) for the first time [64]. They speculated that SATL1 mutations may be a potential factor in the development of late-onset ASD symptoms. The patients showed delayed speech, repetitive movements, a lack of communication, and frequent feelings of anxiety. However, the patients showed no abnormalities in their mental development until they were seven to eight years old. One of the patients, whose parents were consanguineously married, was diagnosed with tonic–clonic epilepsy at the age of one. Interestingly, quite similar to SSAT1, SATL1 has N1 acetyltransferase activity and can bind with spermidine. The discovered SATL1 mutation (Y601*) introduces a termination codon prematurely, which may alter its enzyme activity and consequently cause dysregulation of polyamine metabolism. The detailed mechanism will need further investigation.

Due to the low number of cases, the treatment of polyaminopathy patients has been faced with significant challenges. To date, only BABS patients have a specific drug, DFMO, mostly because DFMO was previously approved to treat African sleeping sickness and has been extensively tested in clinical trials for cancer treatment. The treatment for the remaining four polyaminopathies is mainly symptomatic treatment to alleviate the exacerbation of the diseases. The potential therapeutic agents of SRS and FABAS have achieved significant results in patient cells or yeast models but have not been reported to be tested in humans. Spermine pre-drugs [40], PBA [41], DFMO [43], and Me_2_SPM [42] showed beneficial effects in the cells of SRS patients by rebalancing polyamine homeostasis through an increase in the spermine level or a decrease in the spermidine level. In the FABAS yeast model, spermidine partially restored EIF5A function and the resultant phenotype [23]. Studies have shown spermidine is safe and well tolerated in mice and humans [65], which might make spermidine supplements a therapeutic option for FABAS.

The study of polyaminopathies established the causative connection between polyamine metabolism and diseases. With advances in technologies such as whole-genome and whole-exome sequencing, the detection rate of polyaminopathies might keep increasing. In 2020, the International Center for Polyamine Disorders (ICPD) was established at Michigan State University and Spectrum Health West Michigan to study and treat polyaminopathies. Along with the previous international collaboration organizations in polyamine research, such as the International Polyamines Foundation ONLUS, the study and treatment of polyaminopathies could gain new momentum soon.

## Figures and Tables

**Figure 1 ijms-25-06340-f001:**
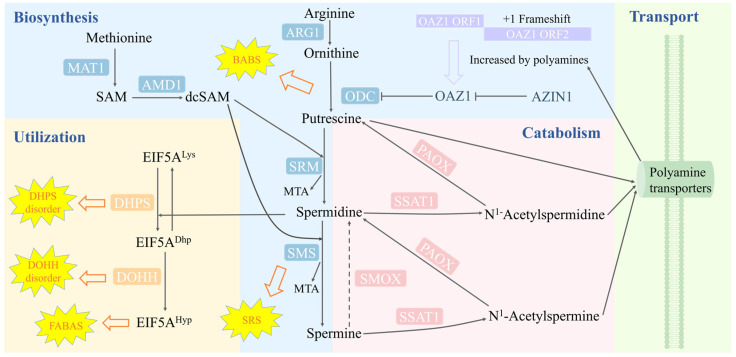
The cellular polyamine homeostasis is maintained by the biosynthesis, catabolism, and transport of polyamines. The hypusination of EIF5A is the best-known utilization of polyamines in cells. The abbreviations in the figure are the following: MAT1 (S-adenosylmethionine synthase 1); AMD1 (S-adenosylmethionine decarboxylase 1); ARG1 (arginase 1); ODC (ornithine decarboxylase); OAZ1 (ODC antizyme 1); AZIN1 (antizyme inhibitor 1); SRM (spermidine synthase); SMS (spermine synthase); MTA (5′-methylthioadenosine); EIF5A (eukaryotic translation initiation factor 5A); DHPS (deoxyhypusine synthase); DOHH (deoxyhypusine hydroxylase); SMOX (spermine oxidase); SSAT1 (spermidine/spermine N1-acetyltransferase); PAOX (polyamine oxidase); BABS (Bachmann–Bupp syndrome); SRS (Snyder–Robinson syndrome); DHPS disorder (deoxyhypusine synthase disorder); DOHH disorder (deoxyhypusine hydroxylase disorder); FABAS (Faundes–Banka syndrome).

**Figure 2 ijms-25-06340-f002:**
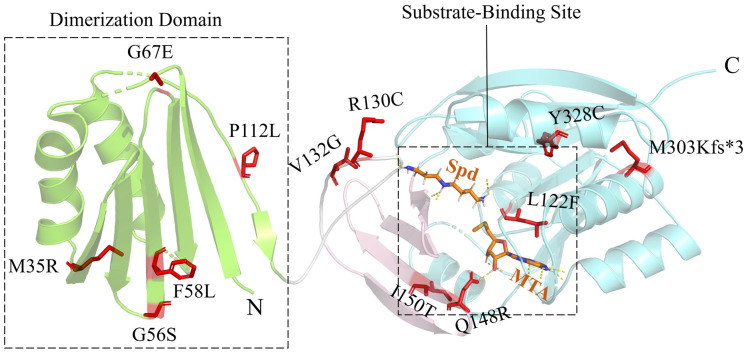
The structure of the SMS protein (PDB: 3C6K). Only a monomer is shown for clarity. The N-terminal, central, and C-terminal domains are colored yellow-green, pink, and light blue, respectively. The loop connecting the N-terminal and central domains is colored white. Spermidine (Spd) and 5′-methylthioadenosine (MTA) bound to SMS are presented as sticks. The mutation sites in SRS patients are shown as red sticks. The asterisk (*) indicates a translation termination codon.

**Figure 3 ijms-25-06340-f003:**
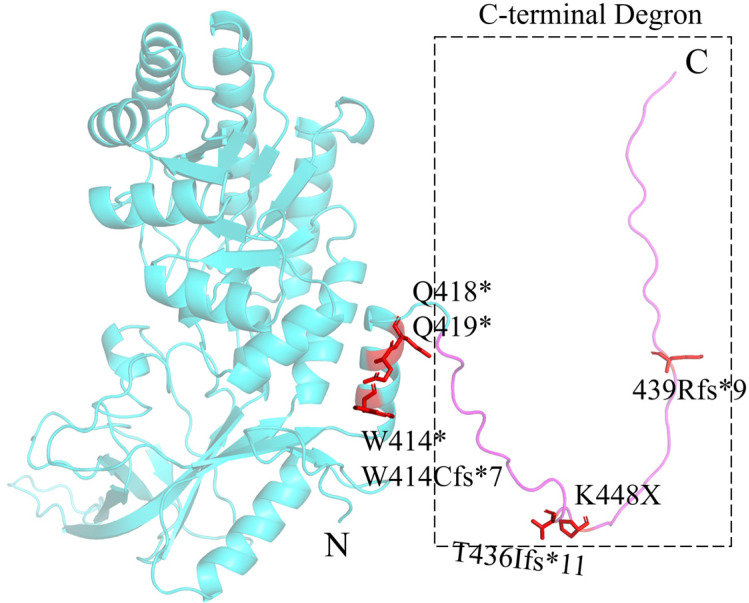
The 3D structure of the full-length human ODC monomer predicted by AlphaFold2. The ODC C-terminal PEST degron is colored magenta and the rest structure is colored light blue. The red sticks are the reported mutation sites in BABS patients, with a star indicating a stop codon.

**Figure 4 ijms-25-06340-f004:**
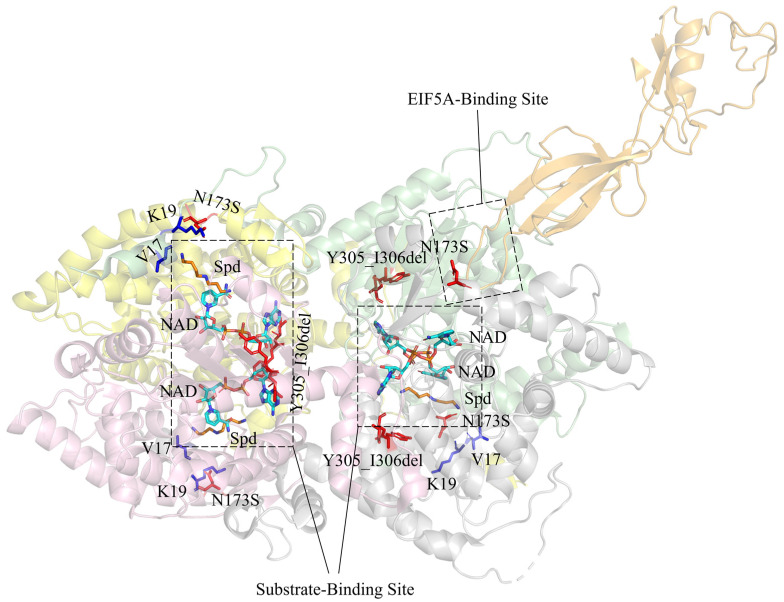
The complex structure of EIF5A-DHPS (PDB: 8A0E). DHPS monomers are colored yellow, pink, green, and grey. EIF5A is colored orange. Spermidine (Spd) and NAD are shown as dark yellow and cyan sticks respectively, with oxygen atoms red and nitrogen atoms blue. The amino acids shown as red sticks are reported mutation sites in DHPS disorder patients. V17 and K19 interact with N173.

**Figure 5 ijms-25-06340-f005:**
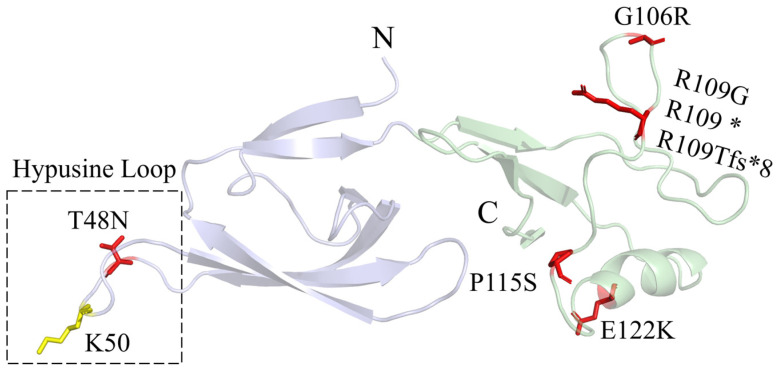
The 3D structure of EIF5A (PDB: 8A0E). The N-terminal and the C-terminal domains are colored blue and green, respectively. The yellow stick is the K50 hypusination site. The red sticks are the reported mutation sites in FABAS patients.

**Figure 6 ijms-25-06340-f006:**
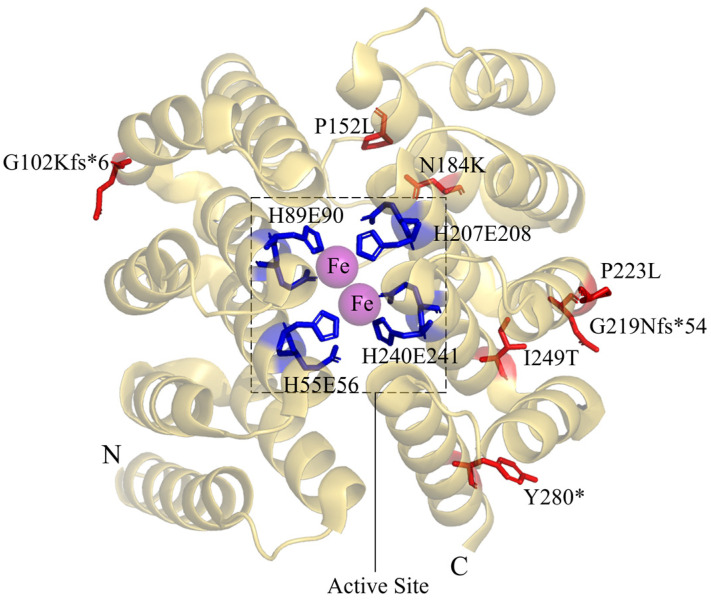
The 3D structure of the DOHH monomer (PDB: 4D4Z). The blue sticks are the four metal coordination sites (HE sites). The red sticks are the reported mutation sites in DOHH disorder patients.

**Table 1 ijms-25-06340-t001:** Genetic modes, patient cases, and clinical symptoms of known polyaminopathies.

Polyaminopathies	Name	Mutant Gene	Genetic Mode	Mutant Form	First Case	Cases	Deaths	Clinical Symptoms
Polyamine biosynthesis-related diseases	SRS	SMS	X-linked recessive	Monogenic mutations	2003	24 males	2 males	Developmental delay, intellectual disability, hypotonia, seizures, osteoporosis, kyphosis, genital abnormalities, facial dysmorphism
	BABS	ODC	autosomal dominant	Monogenic mutations	2018	6 males 5 females	1 male (labor induction)	Developmental delay, intellectual disability, hypotonia, non-congenital alopecia, abnormal brain MRI, non-specific dysmorphic features, macrocephaly
Polyamine utilization-related diseases	DHPS disorder	DHPS	autosomal recessive	Biallelic mutations	2019	1 male 4 females	none	Developmental delay, intellectual disability, seizures, dystonia, pregnancy problems
	FABAS	EIF5A	autosomal dominant	Monogenic mutations	2021	3 males 4 females	none	Developmental delay, intellectual disability, facial deformity, microcephaly
	DOHH disorder	DOHH	autosomal recessive	Biallelic mutations	2022	3 males 2 females	1 male 1 female	Developmental delay, intellectual disability, brain MRI abnormalities, microcephaly, congenital cardiac malformations

**Table 2 ijms-25-06340-t002:** Gene mutations of known polyaminopathies. The asterisk (*) indicates a translation termination codon, and the question marker (?) indicates an unknown mutation.

Polyaminopathies	Name	Mutant Gene	Variants (Gene)			Variants (Protein)		
Polyamine biosynthesis-related diseases	SRS	SMS	c.104T>G c.166G>A c.174T>A c.200G>A c.329+5G>A	c.335C>T c.388C>T c.395T>G c.443A>G c.449T>C	c.831G>T c.908_911del c.983A>G	p.M35R p.G56S p.F58L p.G67E p.?	p.P112L p.R130C p.V132G p.Q148R p.I150T	p.L277F p.M303Kfs*3 p.Y328C
	BABS	ODC	c.1240_1241dupTG c.1241+1G>T c.1242_1263del22	c.1242-2A>G c.1252C>T c.1255C>T	c.1307_1311delinsT c.1342A>T c.1313_1316delCTGT	p.W414Cfs*7 p.? p.W414*	p.? p.Q418* p.Q419*	p.T436Ifs*11 p.K448K p.438Rfs*9
Polyamine utilization-related diseases	DHPS disorder	DHPS	c.1A>G c.518A>G	c.912_917delTTACAT	c.1014+1G>A	p.Met1? p.N173S	p.Y305_I306del	p.?
	FABAS	EIF5A	c.143C>A c.316G>A c.324dupA	c.325C>G c.325C>T c.343C>T	c.364G>A	p.T48N p.G106R p.R109Tfs*8	p.R109G p.R109* p.P115S	p.E122K
	DOHH disorder	DOHH	c.304delG c.455C>T c.552C>A	c.654_655insAACC c.668C>T c.746T>C	c.840T>A	p.G102Kfs*6 p.P152L p.N184K	p.G219Nfs*54 p.P223L p.I249T	p.Y280*

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
