# Peer review of "Structural Insights into the Mechanisms Underlying Polyaminopathies"

_ijms, 2024, doi:10.3390/ijms25126340_

Round 1

Reviewer 1 Report

Comments and Suggestions for Authors

With a great interest I read the paper by Wu and Liu entitled Structural mechanism of the proteins causing polyaminopathies. Overall, I find this topic very important and timely and the Authors choice to review the mechanisms underlying the processes causing polyaminopathies is very good. However, despite the fact of choosing interesting topic I find the manuscript as of moderate quality and the improvement is necessary for the publication recommendation.

Title: I do not understand what kind of Structural mechanism of the proteins are causing polyaminopathies.  The wording structural mechanism is confusing. Maybe Structural insights into the mechanisms underlying polyaminopathies? Please reconsider the title.

Line 42. It is overstatement. I am sure that hypusination is well know, but if it is the best-known? There are plenty of processes polyamines are involved in. Rephrase.

Line 51/52 grammar to correct

Line 63 . This associated-but-not-causal relationship – not sure if this expression is here correct. Rephrase.

Table 1 – “A male with induced labor” – not really proper phrasing

Line 117 - introduces a BsaJI restriction endonuclease site at the gene level, which may interrupt 117 the complete transcription of SMS. Well, I am not sure if human expresses restriction enzymes, yet.

A bit laconic introduction. Especially lines 54-57 can be more elaborative.

Lack of general information on polyamines’ function.

Fig 1. Transporters are named Import/Exports. Are there particular examples of such transporters. Include in the figure.

Line 187 - DHPS is a tetrameric enzyme consisting of four identical subunits. Lacks citation, eg. PMID: 32235505 Lacks info on Spd binding.

Line 197 - p.Y305_I306del leads to the failure of spermidine binding and may also attenuate the bind- 197 ing of NAD. – it does a lot more. It destabilizes protein, DHPS is no longer tetramer but dimer.

Figure 7. – remove, it does not improve anything as such.

328 its International Polyamines Foundation ONLUS not ETS Onlus

Overall, I recommend major revision and then reconsideration. 

Comments on the Quality of English Language

The manuscript needs proofreading/editing. 

Reviewer 2 Report

Comments and Suggestions for Authors

The current manuscript by Wu and Liu represents a timely review of polyaminopathies and the responsible enzymes.   There are a few factual errors or misrepresentations that need to be corrected and some significant areas where editing by a native English speaker would improve the manuscript greatly.  The major points to be correct are listed below.

1)        Figure 1 shows SSAT1 acetylating putrescine.  This reaction is not known to occur in most cell types and the acetylation is performed by a different acetyltransferase.

2)        Page 6, lines 106-108 imply that it is a fact that spermidine accumulation increases toxic aldehyde and ROS production in SRS patients.  Although this is a hypothesis, it has not yet been proven.

3)        Page 6, line 136, the author cited family name is Zhai not Grace.

Comments on the Quality of English Language

There are several passages that require English editing to make them either more correct or more understandable. Generally speaking, the authors have done a good job, but some areas appear to have been missed in the editing process.

Round 2

Reviewer 1 Report

Comments and Suggestions for Authors

Accept. 

Comments on the Quality of English Language

Still reccomend proofreading/check by editorial team.